# Effect of *Candida intermedia* LAMAP1790 Antimicrobial Peptides against Wine-Spoilage Yeasts *Brettanomyces bruxellensis* and *Pichia guilliermondii*

**Rubén Peña**[ID]**, Jeniffer Vílches, Camila G.-Poblete and María Angélica Ganga** *

Laboratorio de Biotecnología y Microbiología Aplicada, Departamento de Ciencia y Tecnología de los Alimentos, Universidad de Santiago de Chile, Santiago 917020, Chile; ruben.pena@usach.cl (R.P.); jeniffernatalia221@gmail.com (J.V.); camila.gonzalezpo@usach.cl (C.G.-P.)
* Correspondence: angelica.ganga@usach.cl; Tel.: +56-2-27184509

**Abstract:** Wine spoilage yeasts are one of the main issues in the winemaking industry, and the control of the *Brettanomyces* and *Pichia* genus is an important goal to reduce economic loses from undesired aromatic profiles. Previous studies have demonstrated that *Candida intermedia* LAMAP1790 produces antimicrobial peptides of molecular mass under 10 kDa with fungicide activity against *Brettanomyces bruxellensis*, without affecting the yeast *Saccharomyces cerevisiae*. So far, it has not been determined whether these peptides show biocontroller effect in this yeast or other spoilage yeasts, such as *Pichia guilliermondii*. In this work, we determined that the exposure of *B. bruxellensis* to the low-mass peptides contained in the culture supernatant of *C. intermedia* LAMAP1790 produces a continuous rise of reactive oxygen species (ROS) in this yeast, without presenting a significant effect on membrane damage. These observations can give an approach to the antifungal mechanism. In addition, we described a fungicide activity of these peptides fraction against two strains of *P. guilliermondii* in a laboratory medium. However, carrying out assays on synthetic must, peptides must show an effect on the growth of *B. bruxellensis*. Moreover, these results can be considered as a start to develop new strategies for the biocontrol of spoilage yeast.

**Keywords:** antimicrobial peptides; *Brettanomyces bruxellensis*; *Candida intermedia*; *Pichia guilliermondii*; reactive oxygen species

## 1. Introduction

Alcoholic fermentation is the process of monosaccharide's conversion to ethanol and $CO_2$. Therefore, the anaerobic metabolism of *Saccharomyces cerevisiae* is the main cause of wine fermentation. However, in spontaneous wine production other yeasts participate such as *Hanseniaspora, Candida, Pichia*, and *Metschnikowia* genera, among others [1]. Nevertheless, these yeasts have lower fermentation capacity, and are not able to grow in high ethanol concentration conditions, given that the *Saccharomyces* genus is the predominant during the final stages of fermentation [1].

Among the unfavorable growth wine conditions, several yeasts are capable to proliferate and generate undesired characteristics in the final product. *Brettanomyces bruxellensis* has been described as the main spoilage yeast during the maturity stage of wine in barrels [2,3]. This yeast has the capacity of transforming hydroxycinnamic acids into vinyl and ethyl derivates, which produce off-flavors in wine [4,5]. Additionally, this aromatic defect can be produced in early stages of fermentation by other yeasts such as *Pichia guilliermondii* [2]. Among these, there are strains which can transform *p*-coumaric acid in 4-vinylphenol in similar proportions as described for *B. bruxellensis*, being that *P. guilliermondii* is a potential problem for winemaking [6]. Because of this, in this industry the use of

sulfites is a widespread strategy to control growth of undesired microorganisms. Nevertheless, several strains are resistant and the use of sulfites in high quantities is potentially unsafe for human health [7]. As a result, alternative strategies such as antimicrobial peptides (AMPs) have been proposed to biocontrol spoilage microorganisms [8–12]. AMPs are low molecular mass peptides with amphipathic characteristic which can affect the growth of several microorganisms by permeabilization of plasmatic membranes and/or by increasing the reactive oxygen species [13–15]. Previously, Peña et al. [16,17] have described antimicrobial peptides production in *Candida intermedia*, which reduce the viability of different *B. bruxellensis* strains in a laboratory medium without affecting the growth of fermentative yeast *S. cerevisiae*. However, it has not yet been described how these peptides affect *B. bruxellensis* and if they are able to inhibit the growth of *P. guilliemondii*. Thus, the aim of this work was to explore the cellular damage produced by *C. intermedia* LAMAP1790 peptides above *B. bruxellensis* and determine if they can control the growth of yeast *B. bruxellensis* and *P. guilliermondii* using laboratory culture mediums and synthetic must. This knowledge will allow to determine the antimicrobial peptides produced for *C. intermedia* as a possible biocontroller in the wine industry.

## 2. Materials and Methods

### 2.1. Strains and Culture Media

The strains of *B. bruxellensis* LAMAP2480, *C. intermedia* LAMAP1790, *Pichia guilliermondii* LAMAP3202, LAMAP3203, *S. cerevisiae* BY4741, and EC1118 were obtained from the culture collection at the Laboratory of Biotechnology and Applied Microbiology, University of Santiago de Chile. *C. intermedia* LAMAP1790 was isolated in Chile from must in the early stages of fermentative process [18] and *B. bruxellensis* LAMAP2480 was isolated from Chilean wine [19]. Both strains of *P. guilliermondii* were isolated from Argentinian vineyards. The strain LAMAP3202 and LAMAP3203 was characterized by Sangorrín et al. (2013), labeled as P7 and P8 strains respectively [20]. All strains used in this work were grown on GYEB media (yeast extract 5 g/L and glucose 20 g/L, adjusted to pH 5.0 with 100 mmol/L phosphate-citrate buffer) [21].

### 2.2. Obtained Supernatant with Antifungal Activity of C. intermedia and Characterization of the Protein Nature of This Activity

To obtain the supernatant with antifungal activity from *C. intermedia* LAMAP1790, the yeast was inoculated in 100 mL GYEB medium during 48 h at 28 °C with orbital agitation at 120 rpm. Then, the culture was centrifuged during 10 min at 5900× *g* to obtain saturated culture supernatant. Afterward, a cut-off of total proteins present in the supernatant was done by means of ultrafiltration in devises *Amicon*® *Ultra-15* with 10 kDa cutoff (Merck-Millipore®, Darmstadt, Germany). In this work, the antifungal supernatant is defined as the fraction obtained from ultrafiltration which only contains proteins of molecular mass under 10 kDa. This antifungal supernatant was sterilized using disposable filters with 0.22 μm pore size (Membrane Solutions LLC®, Windham, NH, USA) and stored at −20 °C to be used later. To determine whether the antifungal activity is related with the presence of peptides with molecular mass under 10 kDa, the antifungal supernatant was treated with 2 mg/mL protease of *Streptomyces griseus* (Sigma-Aldrich®, St. Louis, MO, USA) during 4 h at 37 °C.

### 2.3. Determination of the Cellular Damage Produced on B. bruxellensis by Exposure to Antifungal Supernatant of C. intermedia

The obtained antifungal supernatant of *C. intermedia* LAMAP1790 was assessed to determine if it produces: (a) membrane permeability or (b) rise of the reactive oxygen species (ROS) on exposed *B. bruxellensis* cells during different periods, similar to described by [22] and [23]. Then, $3 \times 10^5$ *B. bruxellensis* LAMAP2480 cells were exposed individually to 1 mL of sterile antifungal supernatant and incubated during 12 h and 24 h at 28 °C. As positive control, a similar number of cells with 600 μg/mL zymolyase 100*T* (Amsbio®, Abingdon, OX, UK) was inoculated at 37 °C during 2 h and

then exposed to 30% $H_2O_2$ during 30 min. As negative control, the same concentration of cells was used and at 28 °C in buffer HEPES saline 1× pH 7.0 (70 mM NaCl, 0.75 mM $Na_2HPO_4$, 25 mM HEPES) were incubated. After treatments, the cells were washed 3 times with buffer HEPES saline 1× pH 7.0. To facilitate the observation, yeast was stained with calcofluor white (Sigma®) in 1:1 proportion with KOH to 10% p/v. The membrane permeability was assessed by means of staining with 2 μM propidium iodide (Sigma®) and the accumulation of ROS was determined by means of staining with 10 μM 6-carboxy-2′,7′-dichlorodihydrofluorescein diacetate (C400; Thermo-Scientific®, Waltham, MA, USA). The fluorescent cells were observed using the epifluorescence microscope Moticam Pro BA410 (Motic®, Xiamen, China), with 40× fluorescence microscope objective lent.

### 2.4. Screening the Antifungal Activity of C. intermedia LAMAP1790 on B. bruxellensis, P. guilliermondii, and S. cerevisiae

The qualitative determination of the antifungal activity of *C. intermedia* LAMAP1790 on *B. bruxellensis* LAMAP2480, *P. guilliermondii* LAMAP3203, LAMAP3203 and *S. cerevisiae* EC1118 strains was carried out following the methodology used by [16]. For this, $1 \times 10^5$ cells from each strain were inoculated in 25 mL warm agar MBA (5 g/L yeast extract, 5 g/L peptone, 20 g/L glucose and 15 g/L agar, adjusted to pH 5.0 with 100 mmol/L phosphate-citrate buffer and supplemented with 0.03 g/L of methylene blue). Each inoculated media was plated into petri dishes. A surface inoculation was carried out using 10 μL of $1 \times 10^8$ cells/mL suspension from *C. intermedia* LAMAP1790. As control, the same surface inoculation of *S. cerevisiae* BY4741 was carried out. The plates were incubated for 7 days at 28 °C and every assay was evaluated six times. The qualitative determination was done by observation and measuring the inhibition halo present in the plates.

### 2.5. Antifungal Activity of Low Mass Peptide Fraction Obtained from C. intermedia Antifungal Supernatant against B. bruxellensis, S. cerevisiae and P. guilliermondii in Synthetic Must

The obtention of 100X concentrated low mass peptide fraction (under 10 kDa) was performed by lyophilization (IlShineBioBase® freeze dryer, Dongducheon-si, Gyeonggi-do, Korea) of 3 L to sterile antifungal supernatant derived from cultures of *C. intermedia* LAMAP1790 in GYEB medium. The total protein quantification in the fraction was done according to [24]. The evaluation of the antifungal activity was done using simultaneous inoculation of *S. cerevisiae* EC1118 and *B. bruxellensis* LAMAP2480 or *P. guilliermondii* LAMAP3202 in synthetic grape must, (100 g/L glucose, 100 g/L fructose, 5 g/L maleic acid, 0.5 g/L citric acid, 3 g/L tartaric acid, 0.75 g/L potassium phosphate, 0.5 g/L potassium sulfate, 0.155 g/L calcium chloride, 0.25 g/L magnesium sulfate, 0.2 g/L sodium chloride, 4 mg/L manganese sulfate, 1.5 mg/L calcium pantenoate, 2 mg/L nicotinic acid, 0.25 mg/L thiamine hydrochloride and 0.003 mg/L biotin; pH 3.5) [25], Previously, each strain was adapted to the media using a procedure described by [26]. To the antifungal assays, 5 mL synthetic must was inoculated with $1 \times 10^2$ cells of *S. cerevisiae* EC1118, *B. bruxellensis* LAMAP2480 or *P. guilliermondii* LAMAP3202 strains individually (determined by direct yeast count in Neubauer chamber), and supplemented with 1 μg of low mass peptide fraction. As a control, the same procedure was carried out, but the medium was supplemented with 1 μg of total proteins obtained from the concentrate sterile culture supernatant of *S. cerevisiae* BY4741 (*IlShineBioBase® freeze dryer*, Dongducheon-si, Gyeonggi-do, Korea). Each assay was incubated for 21 days, and every 3 days a cellular count of the cultures was carried out on YPD agar plates (5 g/L yeast extract, 5 g/L peptone, 20 g/L glucose and 20 g/L agar) incubated for 7 days at 28 °C. The count of spoilage yeast in the mixed culture was performed in YPD agar plates supplemented with 0.01% *v/v* of cycloheximide, according to [27].

### 2.6. Statistical Analysis

All the data was analyzed using the Kruskal–Wallis test, with an initial analysis of the distribution goodness of fit using the Kolmogorov–Smirnov test. All analysis was carried out with Statgraphics

Centurion XVI Software (Statpoint Technologies Inc., Warrenton, VA, USA). The significant differences were validated with a probability < 0.05.

## 3. Results and Discussion

One of the most important aspects in the study of AMPs is to determine its antifungal action mechanism. In relation to antimicrobial peptides to biocontrol contaminant microorganisms in winemaking, Enrique et al. (2008) [9] studied the antifungal effect of the synthetic peptide LfcinB$_{17-31}$ on *B. bruxellensis*, determining that its action mechanism is related to the penetration of the peptides into the cell cytoplasm. Additionally, by fluorescence microscopy, Branco et al. (2017) [12] have described that saccharomycin (antifungal peptides produced by *S. cerevisiae* CCMI885 strain) produce cell membrane disruption and internalization of the peptides in *Hanseniaspora guilliermondii* and *B. bruxellensis*. Our previous results have demonstrated that *C. intermedia* LAMAP1790 releases peptides in the culture medium with masses under 4.6 kDa, which show selective antifungal activity on *B. bruxellensis* strains [16,17]. With the purpose of defining the cell damaged produced by the antifungal supernatant of *C. intermedia* LAMAP1790 (which contains these peptides) on *B. bruxellensis* LAMAP2480, different assays were carried out using calcofluor white (CW), propidium iodide (PI), or 6-carboxy-2',7'-dichlorodihydrofluorescein diacetate (C400) [12,22,28] (Figure 1).

Under optimum growth condition, the yeast wall stains bright blue-white by the CW assembly [28], being impermeable to PI [12] and the C400 cannot be oxidized to its fluorescence form [22]. As a negative control, cells were inoculated in buffer HEPES saline pH 7.0 (Figure 1A–C), it can be observed that neither the wall cellular nor the impermeability of the membrane was affected. Only 29.91 ± 7.29% of the observed cells in the medium show green fluorescence (Figure 1C) and none show red color. This increase of green fluorescence would be related to the lack of nutrients that *B. bruxellensis* had during the 24 h trail, due has been reported that such periods may activate an autophagy process [29,30]. Autophagy is a non-selective degradation of organelles or intracellular macromolecules, a recycling process that allows the amino acid supply and survival. *S. cerevisiae* can do mitophagy (removal of damaged mitochondria), therefore, releasing mitochondrial ROS into the cytoplasm [29,30]. On the other hand, when the cells have damage in the membrane, this is no longer impermeable to PI, dying cells in red [23].

Thus, as a positive control of both processes, we carried out an induction to the oxidative stress and membrane damage by zymolyase and H$_2$O$_2$ treatment (Figure 1D–F). As observed in this figure the treatment produced a 63.39 ± 6.92% permeabilization to cell surface membrane, allowing the penetration of PI into the cell (compared with control sample 1B). Besides, a 63.61 ± 8.17% of cells show a rise of intracellular ROS, which allowed the observation of green fluorescence derived from C400 oxidation (Figure 1F). Additionally, when yeasts were exposed to *C. intermedia* supernatant at 12h, it was observed a rise in the number of cells which oxide C400 (Figure 1I) which is sustained at 24 h of incubation (Figure 1L), while it is observed a little rise of permeable cells of PI to 24 h of incubation (Figure 1H,K). When the *C. intermedia* supernatant is treated with protease, a decrease decrease in the number of cells that oxidize C400 and the permeable cells of PI (Figure 1N,O) was observed, confirming that antifungal compounds have protein nature [16].

By comparing the percentage of fluorescent yeast in different conditions (Figure 2), it can be observed that the incubation of *B. bruxellensis* with the antifungal supernatant produce a sustaining little rise in the number of permeable cells to PI at incubation time, is not statistically different from the negative control (Figure 2A).

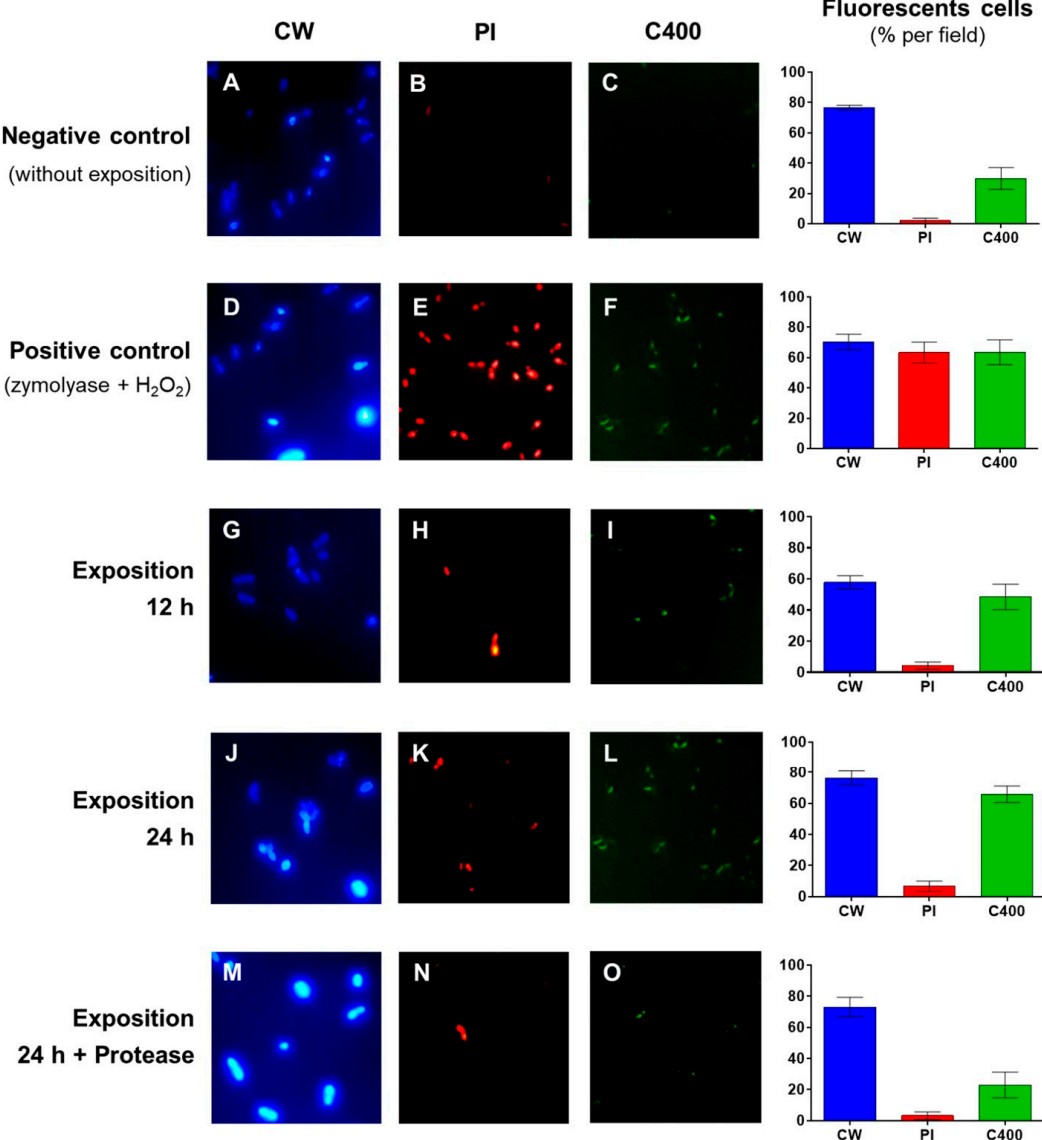

**Figure 1.** Evaluation of permeability and reactive oxygen species (ROS) accumulation in *B. bruxellensis* LAMAP2480 cells exposed to *C. intermedia* LAMAP1790 antifungal supernatant at different times, using epifluorescence microscopy. Graphics at the right side of each line represents a percentage of fluorescent cells per field counted in each treatment. (**A–C**): untreated yeasts (Negative control). (**D–F**): yeasts exposed to $H_2O_2$ 30% *v/v* for 30 min, after treatment with zymolyase (600 μg/mL) for 2 h at 37 °C (Positive control). (**G–I**): yeasts exposed to *C. intermedia* antifungal supernatant for 12 h. (**J–L**): yeasts exposed to *C. intermedia* antifungal supernatant for 24 h. (**M–O**): yeasts exposed to *C. intermedia* antifungal supernatant for 24 h after a proteolytic treatment to the supernatant for 4 h at 37 °C with 2 mg/mL of *Streptomyces griseus* protease (Sigma®). CW: calcofluor white staining, PI: propidium iodide staining, C400: 6-carboxy-2′,7′-dichlorodihydrofluorescein diacetate staining.

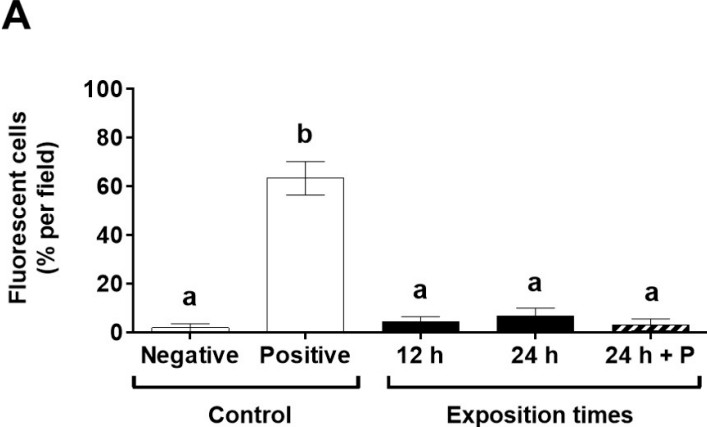

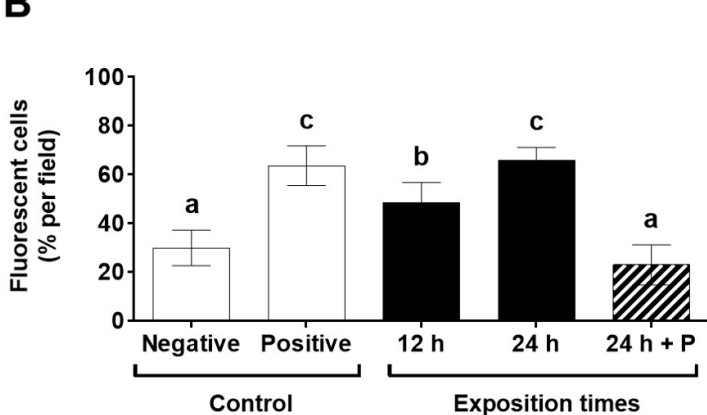

**Figure 2.** Evaluation of damage produced in *B. bruxellensis* cells after 12 h and 24 h of exposition to antifungal supernatant from *C. intermedia* LAMAP1790. (**A**): Membrane damage observed by cell permeability to propidium iodide staining, (**B**): Oxidative damage derived of ROS accumulation measured with 6-carboxy-2″,7′-dichlorodihydrofluorescein diacetate staining. The evaluation was performed comparing the percentage of stained fluorescent cells of *B. bruxellensis* after each treatment. White bars correspond to control treatments, black bars correspond to yeasts exposed to antifungal supernatant at labeled times and bars in striped lines (labeled 24 h + P) correspond to 24 h of yeasts exposition to antifungal supernatant treated previously with 2 mg/mL of *Streptomyces griseus* protease (Sigma®). Different letter above each column represents a significative difference ($p < 0.05$).

Nevertheless, by evaluating the percentage of cells which oxidize C400, they increase significantly as incubation time increases, even achieving similar values to the obtained due to the introduction to oxidative stress with $H_2O_2$ (Figure 2B). These results allow to demonstrate that the sustaining increase of ROS in *B. bruxellensis* is related to the presence of peptides of mass under 10 kDa in the antifungal supernatant of *C. intermedia* LAMAP1790 and propose that its antifungal action mechanism would be related to the oxidative damage that the exposed cell suffers to the supernatant. This should be proved by the non-significant rise of the observed permeability to PI in *B. bruxellensis* between 12 h and 24 h of exposure to antifungal supernatant, because it has been demonstrated that the induction of ROS in yeasts such as *H. guilliermondii* produces the permeabilization of its cellular membrane [11]. Similar effects have been reported to synthetic peptide PAF26 and other similar peptides in which have been demonstrated that they can penetrate the cytoplasm of *S. cerevisiae*, without affecting firstly the integrity of the cellular membrane [13,14,31]. Thus, it determined that the synthetic peptide PAF26 would have a multistep mechanism of action, where it first interacts with the wall or cellular membrane, then it would be endocytosed and accumulated in the vacuoles, and finally, it would be transported to the

cytoplasm and perform its antifungal action [14]. This mechanism would be like the observations made for *B. bruxellensis* by means of fluorescence microscopy.

It was carried out a qualitative antifungal test with *S. cerevisiae*, *B. bruxellensis*, and two strains of *P. guilliermondii* in solid MBA agar plates to determine the formation of inhibition halos produced by *C. intermedia* LAMAP1790. The two strains of *P. guilliermondii* selected was previously studied by Sangorrín et al. (2013) [20]. In that work, from a pool of 15 strains, it was possible to conclude that strains LAMAP3202 and LAMAP3203 (labeled by Sangorrín as P7 and P8) have the highest transformation efficiencies of *p*-coumaric acid in 4-vinylphenol (more aggressive wine-spoilage phenomena). For these reasons, we considered these strains as the best model to our study. As shown in Table 1, *S. cerevisiae* EC1118 strain does not show growth inhibition, while the *B. bruxellensis* LAMAP2480 strain shows a clear inhibition halo surrounding culture of *C. intermedia*, whose diameter reached $19.00 \pm 0.62$ mm, as it was described by Peña et al. [16,17]. By analyzing the behavior of *P. guilliermondii*, LAMAP3202 and LAMAP3203 strains can be observed that an inhibition halo appears, whose diameters reached $15.33 \pm 0.82$ mm and $16.17 \pm 0.75$ mm, respectively (Table 1). Then, *B. bruxellensis* shows a greater sensitivity to the presence of *C. intermedia* than *P. guilliermondii*. Similar studies carried out by Lopes and Sangorrín (2010) [32] have demonstrated that *P. guillermondii* sensitivity depends on the yeast strains to which it is exposed. On the other hand, Villalba et al. (2016) [23] demonstrated that the production of antifungal compounds of protein nature produced by *Torulaspora delbrueckii*, which has a molecular mass above 30 kDa, shows glucanase and chitinase activity. Therefore, the authors conclude that this would be a killer toxin rather than an antimicrobial peptide (AMP). Thus, this work would constitute the first qualitative evidence which shows the sensitivity of *P. guilliermondii* strains to antimicrobial peptides produced by non-*Saccharomyces* yeasts.

**Table 1.** Inhibition halos obtained after the exposure of *C. intermedia* LAMAP1790 against strains of *S. cerevisiae*, *B. bruxellensis* and *Pichia guilliermondii* in MBA medium.

|  |  | *C. intermedia* **LAMAP1790** Inhibition Halo (mm) |
| --- | --- | --- |
| *S. cerevisiae* | EC1118 | [†] ND [a] |
| *B. bruxellensis* | LAMAP2480 | $19.00 \pm 0.62$ [c] |
| *P. guilliermondii* | LAMAP3202 | $15.33 \pm 0.82$ [b] |
|  | LAMAP3203 | $16.17 \pm 0.75$ [b] |

Values with the same superscript letter are not significantly different ($p < 0.05$). [†] ND: Non-Detected.

With the purpose of determining whether the antifungal effect of *C. intermedia* LAMAP1790 is similar in winemaking conditions, it was carried out assays on synthetic must [12]. We decided to use this media to avoid the antimicrobial influence on yeast described to the hydroxycinnamic acids present in the natural grape must (mainly *p*-coumaric and ferulic acid) [33–35]. Thus, the viability of the spoilage strains *B. bruxellensis* LAMAP2480 and *P. guilliermondii* LAMAP3202 were assessed in mixed culture with *S. cerevisiae* EC1118 for 21 days (Figure 3).

The synthetic must was supplemented with 1 μg of low-mass protein fraction obtained from *C. intermedia* supernatant, and then was inoculated using the spoilage yeasts. Posteriorly, it was inoculated with *S. cerevisiae* EC1118 (fermentation starter). As can be seen in Figure 3 (A, B), the growth of *S. cerevisiae* is not affected, demonstrating the harmlessness of the antifungal peptides against this yeast. In the case of the effect on *B. bruxellensis*, it was observed that its growth decreases in one magnitude order of difference compared to the control (3A), while in the case of *P. guilliermondii* L3202, minimal changes between the treatment and control were observed (Figure 3B). Despite having growth inhibition of *P. guilliermondii* in solid medium (Table 1), this effect was not seen in synthetic must, which can be related to a greater concentration of an antifungal compound, possibly requiring a greater concentration for this specie compared to *B. bruxellensis*. To date, there are no previous studies that assess the antifungal capacity of a compound of protein nature (AMP or killer toxin) on the growth of

*P. guilliermondii* in mixed cultures with *S. cerevisiae* in synthetic wine must. Thus, it would be necessary to further study the action of *C. intermedia* peptides in winemaking conditions.

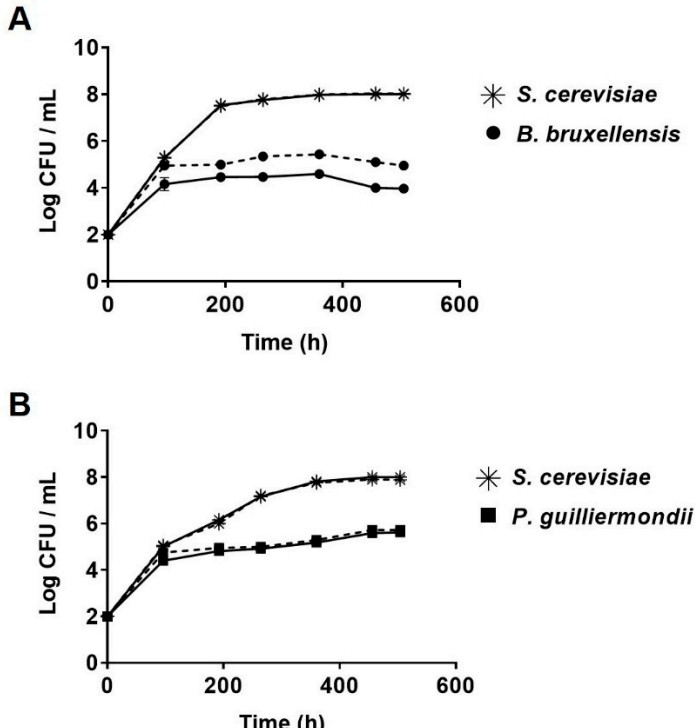

**Figure 3.** Antifungal activity of 1 µg of low mass protein fraction (under 10 kDa) concentrated from *C. intermedia* L1790 supernatant (solid lines) used synthetic wine must. (**A**) *S. cerevisiae* with *B. bruxellensis* L2480 (**B**) *S. cerevisiae* with *P. guilliermondii* 3202. The controls (stripped lines) corresponds to the concentrated supernatant of *S. cerevisiae* BY4741 (not antifungal activity). All assays were performed in triplicate.

## 4. Conclusions

The antifungal supernatant obtained from de culture media of *C. intermedia* LAMAP1790 produces a continuous rise of oxygen reactive species (ROS) in *B. bruxellensis*, without trigger a significant effect on its membrane damage. This effect was totally avoided when the supernatant was treated with a proteolytic enzyme, proving that low mass peptides contained in this fraction are responsible for this effect. Herewith, *C. intermedia* L1790 showed antimicrobial effect on *B. bruxellensis* LAMAP2480, *Pichia guilliermondii* LAMAP3202, and LAMAP3203 when laboratory medium was used; however, similar effect was not observed when synthetic must was used. Therefore, it is necessary to identify the peptides with antifungal activity produced by *C. intermedia* LAMAP1790 and study how some enological factors (pH, ethanol, sugars, etc.) may affect their antifungal capacity. This will allow us to determine its possible technological application in the control of yeast contaminants in the wine industry.

**Author Contributions:** Conceptualization: R.P. and M.A.G.; Methodology: R.P. and M.A.G.; Formal analysis: R.P. and J.V.; Data analysis: R.P., J.V., and C.G.-P.; Supervision: M.A.G.; Writing-Original Draft: R.P., C.G.-P. and M.A.G.; Project Administration: M.A.G. All authors have read and agreed to the published version of the manuscript.

**Funding:** Rubén Peña was funded by the Comisión Nacional de Investigación Científica y Tecnológica CONICYT-PCHA/DoctoradoNacional/2013-21130439 Doctoral Fellowship. Camila G-Poblete was funded by Facultad Tecnológica Fellowship from Universidad de Santiago de Chile.

**Conflicts of Interest:** The authors hereby declare they do not have any conflict of interest associated to this work.

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
