# Peer review of "Effect of Candida intermedia LAMAP1790 Antimicrobial Peptides against Wine-Spoilage Yeasts Brettanomyces bruxellensis and Pichia guilliermondii"

_fermentation, doi:10.3390/fermentation6030065_

Round 1
Reviewer 1 Report
The MS is well-written and experiments are well-designed. I have some questions about the methodology.
Line 59-60: Were the used yeasts isolated from must or wine?
Line 112-115: Why was a synthetic and not a real grape must used in your investigation?
According to the spoilage action of the yeasts Brettanomyces and Pichia they require different medium for optimal growth. I recommned for future investigations an ethanol containing medium or real wine for Brettanomyces strains.
Unfortunatelly on Figure 1 the fluorescent cells are hardly visible. Please try to improve the quality of these pictures.
Author Response
Reviewer 1: Comments and Suggestions for Authors
The MS is well-written and experiments are well-designed. I have some questions about the methodology.
Line 59-60: Were the used yeasts isolated from must or wine?
All the non-Sacharomyces strains used in this work were isolated from wine-related environment. Candida intermedia LAMAP1790 and both strains of Pichia guilliermondii were isolated from must in fermentation process (Ganga & Martínez, 2004 (J. Appl. Microbiol 96:76-83); Sangorrin et al. 2011 (J.Appl. Micrrobiol 114:1066-1074)) and B. bruxellensis LAMAP2480 was isolated from wine (Valdes et al, 2014 (Fems Microbiol.Lett. 361:104-106)). We include this information to manuscript in lines 61-65.
Line 112-115: Why was a synthetic and not a real grape must used in your investigation?
We decided to use synthetic must instead natural grape must to avoid the toxic effect described in yeast by grape natural hydroxycinnamic acids (p-coumaric, caffeic and ferulic acids). So, the presence of these acids in the experiment proposed in this work may lead to errors, enhancing the antifungal activities of the peptides identified in C. intermedia LAMAP1790 (Figure 3). Moreover, for further investigation we will considerate this suggestion. This comment was added at the manuscript in lines 236-238.
According to the spoilage action of the yeasts Brettanomyces and Pichia they require different medium for optimal growth. I recommend for future investigations an ethanol containing medium or real wine for Brettanomyces strains.
We agreed with your suggestion. Several studied in Brettanomyces has been performed with presence of ethanol in the media. However, in this approach we studied the growth inhibitory effect of peptides obtained from C. intermedia LAMAP1790, simulating a mixed culture between S. cerevisiae and spoilages strain at the earlier stages of the fermentation process. To further investigation we will consider how some enological factors (pH, ethanol, sugars, etc.) may affect the antifungal capacity of C. intermedia (lines 275-277).
Unfortunatelly on Figure 1 the fluorescent cells are hardly visible. Please try to improve the quality of these pictures.
In agreed with your suggestion, we improve the visualization of fluorescent cells in Figure 1.
Reviewer 2 Report
The manuscript written by Pena et al. focuses on the antifungal action of a peptide produced by Candida intermedia LAMAP1790. The biocontrol of spoilage microorganism in the wine industry is an actual topic, especially in these years, thus many searches are aimed to find alternative solutions to the use of SO2.
The authors have already published the ability of Candida LAMAP 1790 to produce AMP: Pena et al. 2019 Fermentation, Pena and Ganga Antonie van Leeuwenhoek 112(3) 2018; a poster to the 28th International Conference on Yeast Genetics and Molecular Biology (ICYGMB) and the XXI Congreso Cileno de Ciencias Y Tecnologia De Alimentos Sochital 2017. Therefore, the manuscript lacks originality; the only new knowledge is the effect of these peptides on Pichia guilliermondii and the production of ROS in Brettanomyces.
In general, the paper is well written and well discussed, but the main criticism is due to the low number of species and strains tested. Moreover, the cellular damage should be tested also in P. guilliermondii.
Minor observations:
-lines 59-60: Have the authors verified that P. guilliermondii LAMAP3202 and LAMAP3203 are different strains?
-lines 107-109: Which volume of medium has been lyophilized?
-lines 117-118: How do you estimate the microbial load? OD?
- since in synthetic medium the effect of the antifungal peptide was not observed for Pichia, why didn’t the authors test a higher concentration?
For these reasons, in my opinion, this work needs major revisions.
Author Response
Reviewer 2: Comments and Suggestions for Authors
The manuscript written by Pena et al. focuses on the antifungal action of a peptide produced by Candida intermedia LAMAP1790. The biocontrol of spoilage microorganism in the wine industry is an actual topic, especially in these years, thus many searches are aimed to find alternative solutions to the use of SO2.
The authors have already published the ability of Candida LAMAP 1790 to produce AMP: Pena et al. 2019 Fermentation, Pena and Ganga Antonie van Leeuwenhoek 112(3) 2018; a poster to the 28th International Conference on Yeast Genetics and Molecular Biology (ICYGMB) and the XXI Congreso Chileno de Ciencias y Tecnologia De Alimentos Sochital 2017. Therefore, the manuscript lacks originality; the only new knowledge is the effect of these peptides on Pichia guilliermondii and the production of ROS in Brettanomyces.
In attention to your commentary, the work presented at SOCHITAL (2017) was focused in the description of the antifungal activity of C. intermedia LAMAP1790 against several strains of B. bruxellensis in laboratory media, and the determination of the chemical nature of the compound. In ICYGBM (2017), it was presented studies about the sequentiation of a low molecular weight peptide present in C. intemedia LAMAP1790 culture supernatant, and in-silico assays about its tridimensional structure and biochemical characteristics. Some of these results gave origin to the article published by Peña et al (2018 (Antoine Van Leeuwenhoek 112: 297-304)). In the case of Peña & Ganga (2019 (Fermentation 5: 25)), that correspond a review of the knowledge about chemical and biological approaches to the control of B. bruxellensis growth in different matrix. In that review, it was described results published by Peña et al (2018 (Antoine Van Leeuwenhoek 112: 297-304)) and other complementary results, to support the biotechnological potential of C. intermedia LAMAP2480 as a bio-control tool.
In the other prospective, the actual manuscript is focused in two major topics. First, we study how the low molecular peptides contained in the culture supernatant of C. intermedia LAMAP1790 exert their antifungal action against B. bruxellensis, using the strain LAMAP2480 as a model. The second goal was studying the antifungal action of the low molecular weight peptides against B. bruxellesis, S. cerevisiae and P. guilliermondii in mixed culture model in synthetic must. This is the first work who study these effects, and all the results showed in this work are novelty described.
In general, the paper is well written and well discussed, but the main criticism is due to the low number of species and strains tested. Moreover, the cellular damage should be tested also in P. guilliermondii.
Regarding in the number of strains used in this work, we can comment that the strain B. bruxellensis LAMAP2480 was selected as a model to study the antifungal mechanism of the C. intermedia supernatant. This was because, in our previous works we have been demonstrated the antifungal capacity in several strains of this spoilage yeast (Peña et al, 2018; Peña & Ganga, 2019). In the case of P. guilliermondii, the two strains used were characterized previously by Sangorrín et al (2013). In that work, from a pool of 15 strain, it was possible conclude that LAMAP3202 and LAMAP3203 (labeled by Sangorrín as P7 and P8) have the highest transformation efficiencies of p-coumaric acid into 4-vinylphenol (discussed in this work on lines 218-222). For this reason, we considered these strains as the best models to our study. However, in further works we hope to expand the number of P. guilliermondii strains and study the cellular damage produced by the low molecular weight peptides produced by C. intermedia LAMAP1790.
Minor observations:
-lines 59-60: Have the authors verified that P. guilliermondii LAMAP3202 and LAMAP3203 are different strains?
Both strains of Pichia guilliermondii was identified as different strains by Sangorrín et al (2013), using RAPD-PCR molecular characterization. We provide additional information of these strains in lines 61-65.
-lines 107-109: Which volume of medium has been lyophilized?
The obtention of 100X concentrated low mass peptide fraction was performed by lyophilization of 3 L to sterile antifungal supernatant derived from cultures of C. intermedia LAMAP1790 (lines 111-113)
-lines 117-118: How do you estimate the microbial load? OD?
Lines 117-118 (now 121-122) describe the initial inoculation in the antifungal assay (point 2.5). We realized a direct count of yeast in Neubauer chamber to estimate de quantity of starter culture needed. This information was added at the manuscript in lines 121-123.
- since in synthetic medium the effect of the antifungal peptide was not observed for Pichia, why didn’t the authors test a higher concentration?
Agreeing with your suggestion, after a 100X concentration of the antifungal supernatant obtained from the culture of C. intermedia LAMAP1790, we obtained a low concentration of peptides The adding the volume needed to supplement with 1 µg, the final volume of the mixed culture varied around 15%. We use that amount to avoid the excessively dilution of synthetic must. We are preparing more material for future studies and will considered your suggestion to study a greater range of concentrations.
Finally, we check the english language and style.
Round 2
Reviewer 2 Report
Dear Editor and Authors
The manuscript has been revised and enhanced in some parts; all the answers are convincing. However, the main criticism of the previous version still remains.
In my opinion, only two tested species are not sufficient; this could be suitable for a research note. The manuscript needs this improvement.
Best regards